# Origin of the Laurentian Great Lakes fish fauna through upward adaptive radiation cascade prior to the Last Glacial Maximum
Nathan J. C. Backenstose [1] ✉, Daniel J. MacGuigan [1], Christopher A. Osborne[1], Moisés A. Bernal[1,9], Elizabeth K. Thomas [2], Eric Normandeau [3], Daniel L. Yule[4], Wendylee Stott[5], Amanda S. Ackiss [6], Victor A. Albert [1], Louis Bernatchez [7,10] & Trevor J. Krabbenhoft [1,8] ✉

The evolutionary histories of adaptive radiations can be marked by dramatic demographic fluctuations. However, the demographic histories of ecologically-linked co-diversifying lineages remain understudied. The Laurentian Great Lakes provide a unique system of two such radiations that are dispersed across depth gradients with a predator-prey relationship. We show that the North American *Coregonus* species complex ("ciscoes") radiated rapidly prior to the Last Glacial Maximum (80–90 kya), a globally warm period, followed by rapid expansion in population size. Similar patterns of demographic expansion were observed in the predator species, Lake Charr (*Salvelinus namaycush*), following a brief time lag, which we hypothesize was driven by predator-prey dynamics. Diversification of prey into deep water created ecological opportunities for the predators, facilitating their demographic expansion, which is consistent with an upward adaptive radiation cascade. This study provides a new timeline and environmental context for the origin of the Laurentian Great Lakes fish fauna, and firmly establishes this system as drivers of ecological diversification and rapid speciation through cyclical glaciation.

The rapid evolution of phenotypically and ecologically diverse species from a common ancestor, known as evolutionary radiations, are often fueled by adaptation to ecological opportunities that include colonization of new habitats and extinction of interspecific competitors[1–3]. There are cases in aquatic systems where multiple sympatric lineages have each radiated into several species or morphs. Species within these sympatric lineages can be linked through spatial and trophic interactions such as overlapping habitat preferences and predator-prey dynamics[4]. Several famous examples of predator-driven prey diversification have been documented[5,6]. Conversely, it has also been theorized that rapid diversification of prey species would likely facilitate similar diversification in its predators, triggering a so called "upward adaptive radiation cascade", although few empirical examples of such instances have been documented[7]. Additionally, there is a scarcity of studies that examine the historical demographic patterns of rapid co-diversification between ecologically-linked lineages[8]. Investigating multiple

sympatric lineages in the early stages of diversification can provide a richer understanding of the mechanistic drivers of speciation by characterizing the impact of the interactions among demographic processes, genetic change, and ecological differences[9].

A well-suited system for exploring the evolutionary consequences of demographic shifts are the native salmonids of the Laurentian Great Lakes (hereafter, Great Lakes), which have diversified geologically recently in conjunction with changes in their environment[10]. Historically endemic to the Northern Hemisphere, salmonid populations were influenced by Pleistocene glaciation cycles resulting in periods of isolation followed by population expansion, diversification, and range overlap[11,12]. One lineage that experienced such periods of isolation and population expansion is the genus *Coregonus* (Salmonidae: Coregoninae; "ciscoes"). Representatives of this most species-rich genus within Salmonidae are dispersed across cold freshwater habitats of Asia, Europe, and North America and are associated

[1]Department of Biological Sciences, University at Buffalo, Buffalo, NY, USA. [2]Department of Geology, University at Buffalo, Buffalo, NY, USA. [3]Plateforme de bio-informatique de l'IBIS (Institut de Biologie Intégrative et des Systèmes), Université Laval, Québec G1V 0A6, Canada. [4]US Geological Survey, Lake Superior Biological Station, Great Lakes Science Center, Ashland, WI, USA. [5]Freshwater Institute, Fisheries and Oceans Canada, 501 University Crescent, Winnipeg, MB R3T 2N6, Canada. [6]US Geological Survey, Great Lakes Science Center, Ann Arbor, MI, USA. [7]Institut de Biologie Intégrative et des Systèmes (IBIS), Université Laval, Quebec City, QC, Canada. [8]RENEW Institute, University at Buffalo, Buffalo, NY, USA. [9]Present address: Department of Biological Sciences, Auburn University, Auburn, AL, USA. [10]Deceased: Louis Bernatchez ✉e-mail: njbackenstose13@gmail.com; tkrabben@buffalo.edu

with a history of taxonomic uncertainty[13–15]. In North America, members of the *Coregonus artedi* species complex exhibit a vast range of ecological, phenotypic, and functional diversity accompanied by little-to-no mitochondrial genetic differentiation among species[16–18], likely resulting from their recent, rapid diversification in lacustrine habitats. Depth gradients within these habitats allow for diversification within the species complex through the niche partitioning within a lake system[19]. These patterns are exemplified in Lake Superior *Coregonus* species such as *Coregonus artedi*, which favor epipelagic strata (5–90 m), while benthopelagic-oriented species such as *C. hoyi* and *C. kiyi* prefer deeper hypolimnetic zones of 55–90 m and >90 m, respectively[20–22] (Fig. 1a). Lake Superior coregonines exhibit adaptive genetic differences across depth gradients, such as spectral tuning of the visual signal transduction protein *rhodopsin*[23,24].

One of the primary native predators of *Coregonus* is Lake Charr (*Salvelinus namaycush*). Like the coregonines, *S. namaycush* is comprised of several ecologically, genetically, and phenotypically divergent morphs (in this case formally recognized as sub-specific 'strains'). These morphs are distributed across depth gradients similar to *Coregonus*[25–28] (Fig. 1a). Within Lake Superior, the "lean" morph is associated with a shallower habitat preference ( < 80 m), whereas the "siscowet" morph has a deeper preferred habitat ( > 80 m)[26,29,30] (Fig. 1a). The deeper-oriented siscowet *S. namaycush* morph is more abundant and has been shown to often feed on deepwater coregonines, such as *C. hoyi* and *C. kiyi*[29–31]. Depth gradients also facilitate presumably adaptive genetic and physiological divergence of siscowet from its shallow-water ancestors, such as morph-specific immune responses[32] and greater foraging capability in low light conditions[33].

The Great Lakes geologic history has been volatile, which would be reflected in the demographic history of fishes in the region. Fluctuations in population size through time are shaped by ecological and evolutionary forces, with the signatures of these influences catalogued in the genomes of extant organisms. Coalescent-based methods can utilize these signatures for estimating changes in effective population size ($N_e$) over geologic time using one or more genomes. Notably, the pairwise sequentially Markovian coalescent model (PSMC)[34] and sequential Markov coalescent + plenty of unlabeled samples (SMC++)[35] are powerful tools for inferring $N_e$ of populations using whole-genome sequence data[36–38]. Understanding how $N_e$ changes through time can help resolve the evolutionary history of species and show how organisms have responded to shifting biotic and abiotic factors in their environment[39]. Moreover, responses to shifting environments by different species can help predict how current and future environmental changes will affect demography. Coalescent-based approaches can also provide a temporal context to speciation events, including radiations that evolve rapidly through the exploitation of ecological opportunities. PSMC and other sequential Markovian coalescent (SMC) models have been used to infer divergence times between species and among intraspecific ecotypes[34,40–43]. The divergence between lineages can be inferred by identifying the point at which divergent $N_e$ trajectories coalesce into a shared population history[43]. When coupled with environmental data, these approaches can contextualize the evolutionary origins of biodiversity by showing how populations responded to environmental forces during the time of divergence.

Here we used genome-wide coalescent-based tools to explore the parallel adaptive radiations of the *C. artedi* species complex and the *S. namaycush* morphs. Because no genome assemblies are available for any of the *Coregonus* study species, we generated a high-quality de novo assembly of the *C. artedi* genome and combined coalescent-based tools with population-level resequencing data to estimate the timing of population size changes in both predator and prey species throughout the Pleistocene. $N_e$

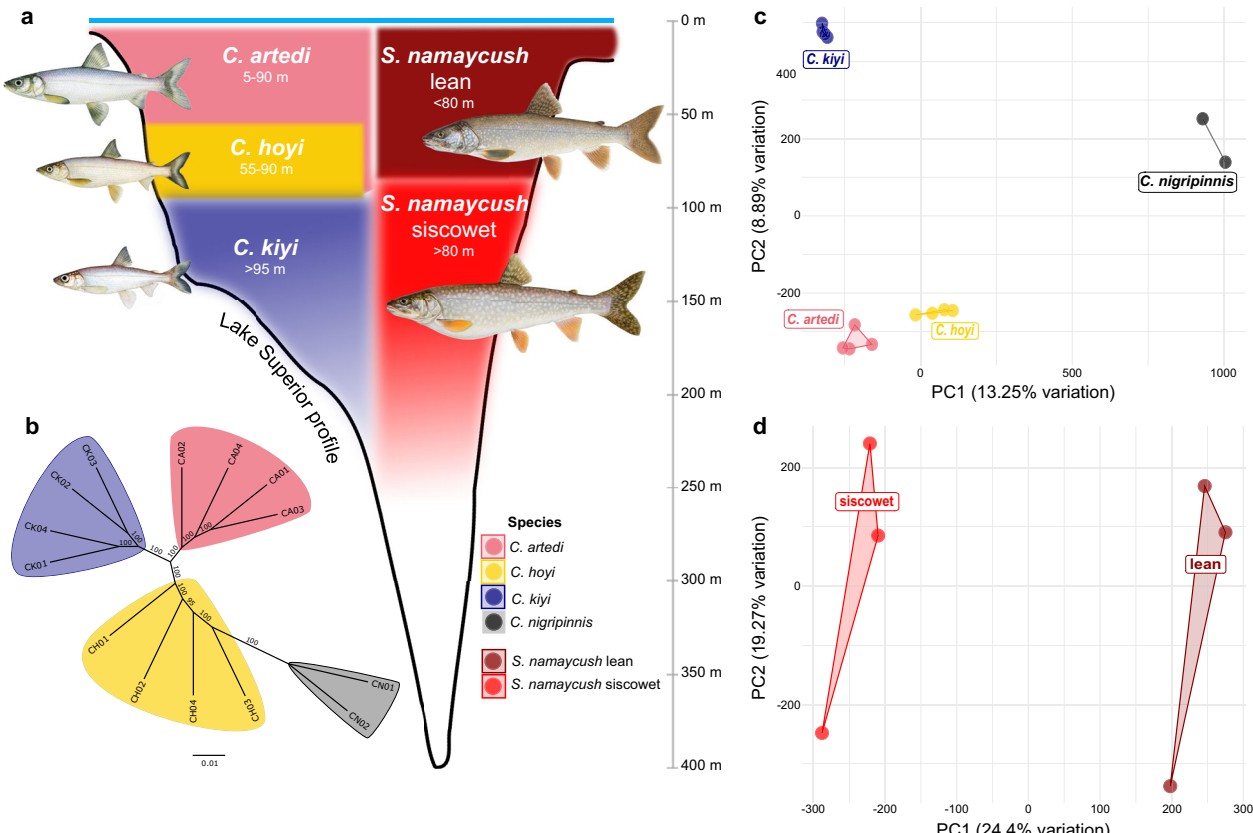

**Fig. 1 | Ecological variation and population structure in Great Lakes salmonids.** **a** Depth profile of Lake Superior showing bathymetric distributions of *Coregonus artedi*, *C. hoyi*, *C. kiyi*, and two morphs of *Salvelinus namaycush* (lean and siscowet). Artwork provided by Joseph R. Tomelleri. **b** Maximum likelihood phylogenetic tree with branch support calculated through 1,000 ultrafast bootstrap replicates. PCA of genetic variation in (**c**) *C. artedi*, *C. hoyi*, *C. kiyi*, *C. nigripinnis* and (**d**) *S. namaycush* lean and siscowet derived from single nucleotide polymorphisms with no missing data. Source data are available at https://doi.org/10.5061/dryad.n02v6wx59.

paleohistories for three *Coregonus* species (*C. artedi, C. hoyi, C, kiyi*) and two morphs of *S. namaycush* (lean and siscowet) from Lake Superior and one *Coregonus* species from Lake Nipigon (*C. nigripinnis*) were estimated to reconstruct demographic histories in the context of environmental change. Previous studies proposed that the *C. artedi* species complex radiated shortly before[44] or following the Last Glacial Maximum (LGM)[45,46]. We used genome-wide data to test these hypotheses and evaluate whether diversification was stimulated by pre-LGM Great Lakes lentic (still water) habitats. Furthermore, comparing $N_e$ trajectories throughout the Pleistocene among Great Lakes salmonid lineages provides insights into the possible co-evolution of a native predator-prey complex, both having likely undergone adaptive radiations in concert. $N_e$ trajectories provide information on the temporal origins of demographic independence among members of the *C. artedi* species complex. Moreover, the evolutionary success of the primary predator species, *S. namaycush*, appears to be contingent on the diversification and abundance of the prey species. Taken together, we identify the geologic context of evolutionary diversification within these lineages with pre-LGM divergence to reflect previously unoccupied niche space in the precursors to the Great Lakes basins of today.

## Results

### *Coregonus artedi* reference genome assembly

The reference genome assembly for *C. artedi* was sequenced with DNA extracted from muscle tissue of an adult female collected in Lake Huron. A total of 11,393,306 Oxford Nanopore sequences ( ~ 28x coverage) that passed quality filtering were generated, resulting in a read N50 of 21,275 base pairs (bp). Illumina (San Diego, CA, USA) DNA sequencing produced an additional 119.9 Gbp ( ~ 40x coverage) of paired-end 150 bp sequence data. An initial raw genome assembly from Flye was 3,037,857,332 bp in length contained within 29,378 contigs. Hi-C scaffolding of the assembled contigs produced well-defined genomic blocks that were then incorporated into a linkage map[47]. Scaffolding of the genome with Chromonomer incorporated ~98% of the polished assembly into 38 linkage groups (chromosomes). The final assembly contained 38 scaffolds and 1,680 unincorporated contigs comprising a total length of 2,491,861,367 bp with a scaffold N50 of 52,236,198 bp and 97.1% complete BUSCOs (Table 1). Repetitive elements comprised a large proportion of the assembly (70.17%) totaling 1.748 Gbp (Supplementary Table 1). Retroelements and DNA transposons comprised 16.53% and 16.28% of the assembly, respectively (Supplementary Table 1). A large percentage of repetitive sequences in the assembly were uncategorized (37.34% of the genome). The dominant repeat families for retroelements and DNA transposons were L2/CR1/REX and Tc1-IS630-Pogo respectively (Supplementary Table 1).

### *Coregonus* species population genomic structure

DNA was extracted from fourteen samples: four each of *C. artedi, C. hoyi*, and *C. kiyi* from Lake Superior and two samples of *C. nigripinnis* from Lake Nipigon. Short-read genome sequencing of these samples produced 11.63–15.38x coverage (Supplementary Table 2). The proportion of reads that were mapped to the reference genome was consistently high across all individuals (99.32–99.89%) with a proper pair alignment range of 94.93–96.1% (Supplementary Table 2). A total of 15,331,196 SNPs were identified. The first PC axis of SNP data ( ~ 13% of the variation) separated *C. nigripinnis* from *C. artedi* and *C. kiyi*, with *C. hoyi* occupying an intermediate position along this axis. PC2 (9% of the variation) separated *C. kiyi* from *C. hoyi* and *C. artedi*, with *C. nigripinnis* being intermediate (Fig. 1c). The third PC axis (8% of the variation) separated *C. hoyi* from the other species. Additional PC axes revealed only individual-level variation (Supplementary Fig. 1). The SNP matrix used for phylogenetic analysis contained 7,008,815 sites after filtering, 60% of which were parsimony informative. TVM + F + ASC + R2 was selected as the best-fit substitution model based on BIC scores. The resulting phylogenetic tree had 100% bootstrap support for all but one branch (Fig. 1b). *C. artedi, C. kiyi*, and *C. nigripinnis* were resolved as monophyletic groups, while *C. hoyi* formed a paraphyletic grade. The longest internal branch in the phylogeny separated

**Table 1 | *Coregonus artedi* genome assembly summary statistics**

| | | |
|---|---|---|
| Initial assembly | Number of contigs | 5,355 |
| | Longest contig (Mb) | 15.18 |
| | Contig N50 (Mb) | 1.78 |
| Final assembly | Length (Gb) | 2.491 |
| | Number of scaffolds | 38 |
| | Number of unplaced contigs [Kb] | 1680 [137] |
| | Scaffold N50 (Mb) | 62.61 |
| | N (%) | 0.03 |
| | GC (%) | 44.57 |
| Assembly BUSCO v5 | Complete (%) | 97.1 |
| (actinopterygii_odb10) | Complete single copy (%) | 55.5 |
| | Complete duplicated (%) | 41.6 |
| | Fragmented (%) | 1.4 |
| | Missing (%) | 1.5 |

*C. nigripinnis* from the Lake Superior species, consistent with the PCA results.

### Demographic history of *Coregonus* species

Demographic reconstructions using PSMC (Fig. 2, Supplementary Fig. 2a,c,e,g) showed that the four species of *Coregonus* had nearly identical $N_e$ trajectories from three million years ago (mya) to approximately 80 thousand years ago (kya). All $N_e$ trajectories declined steadily while remaining in tight correspondence with multiple plateaus occurring through time. Between 80–90 kya, divergence of $N_e$ trajectories occurred and was followed by all species having their lowest $N_e$ estimates during the time of the LGM (19.5–33 kya). At this point in time, *C. artedi* had the lowest estimated $N_e$ for any species (Supplementary Fig. 2a). Following the LGM, all species showed a massive, rapid population expansion (Fig. 2).

Estimates of effective population size through time with SMC++ (Supplementary Fig. 3) began around four mya with consistent $N_e$ trajectories for all species until 60 kya. There was a downward trend in $N_e$ until all species reached their lowest $N_e$ between 30 and 50 kya. The dates for minimum $N_e$ from SMC++ were consistently older than the lowest $N_e$ point inferred from PSMC (Supplementary Fig. 2b,d,f,h, Supplementary Fig. 3). Following the lowest point in $N_e$, SMC++ curves showed rapid $N_e$ increases in all species with peaks appearing at distinct time intervals across species. After the $N_e$ peak seen in the *C. artedi* samples, a split in bootstrap trajectories was apparent that was maintained until the most recent estimates. The most contemporary estimates available through SMC++ ( ~ 100 years ago) showed *C. artedi* having the greatest $N_e$ of ~150,000, followed by *C. hoyi* at ~100,000 and *C. kiyi and C. nigripinnis* with the lowest $N_e$ of ~60,000. Mutation rate sensitivity analyses indicated that decreasing the mutations/site/generation to the lower bound of the confidence intervals for salmonid species presented by Bergeron et al.[48] introduced multiple peaks in $N_e$ estimates between 30–500 kya (Supplementary Fig. 4).

### Demographic history of *Salvelinus namaycush*

To understand population structure and estimate effective population sizes through time within the predator species, three lean and three siscowet *S. namaycush* samples were sequenced with Illumina short-reads. The samples ranged from 16.30–20.45x average alignment depth against the *S. namaycush* reference genome[49] (Supplementary Table 3). The percentage of total reads that aligned to the reference ranged from 97.07–97.39% with a proper pair alignment range of 92.68–93.97% (Supplementary Table 3). Variant calling for the two morphs resulted in 3,534,068 SNPs. Lean and siscowet morphs were strongly separated along PC1 of the SNP data (Fig. 1c, Supplementary Fig. 5).

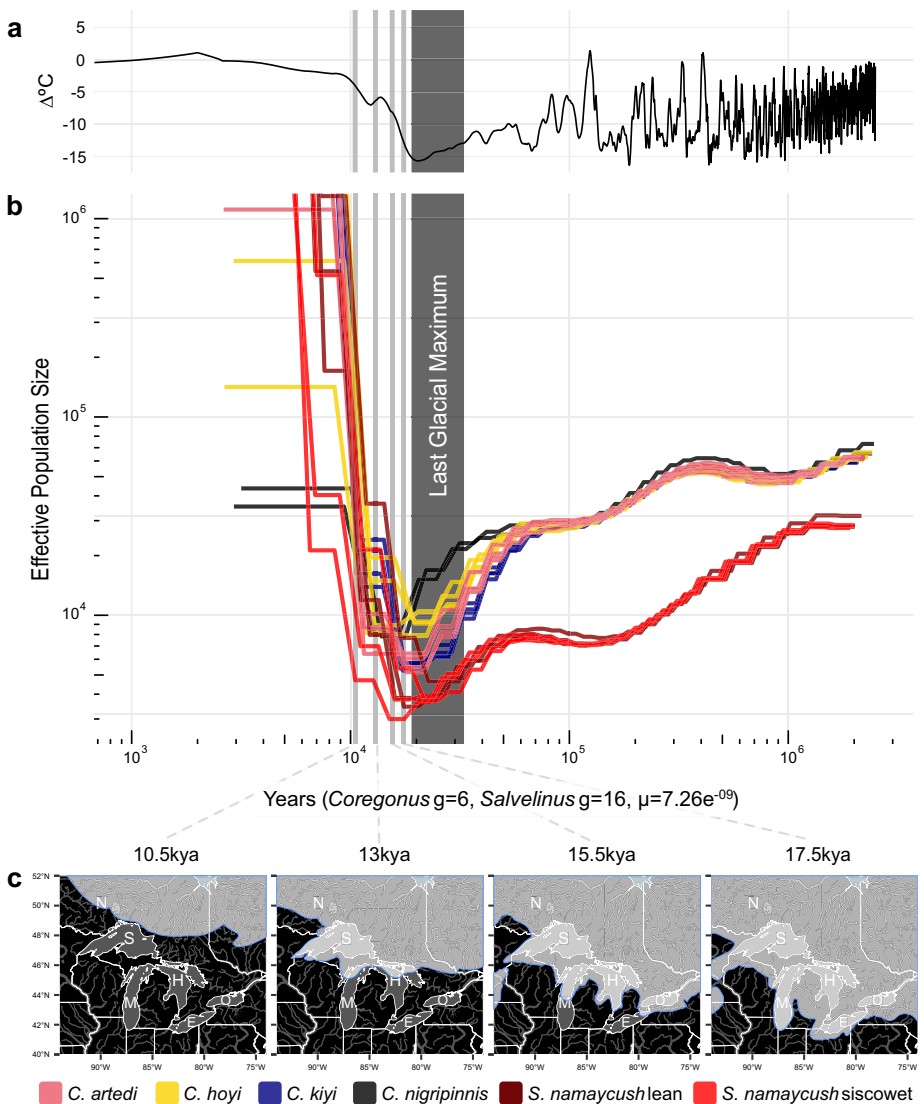

**Fig. 2 | Environmental context of diversification and historical demography of Great Lakes salmonids. a** Mean surface-air temperature of the continental Northern Hemisphere through time, relative to present day. Temperature data from De Boer, Lourens, and Van De Wal[102]. **b** Estimates of effective population size through time via PSMC analysis for four species of *Coregonus* (*C. artedi*, *C. hoyi*, *C. kiyi*, *C. nigripinnis*) and *Salvelinus namaycush* (lean and siscowet). Dark grey bar indicates the time range of the Last Glacial Maximum (19–33 kya). Light grey bars indicate time points (10.5, 13, 15.5, 17.5 kya) of deglaciation stages (**c**) of the Laurentian Great Lakes basin. Shapefiles for glacial margins as well as river and lake boundaries obtained from Dalton et al.[73] and https://www.naturalearthdata.com respectively. Source data are available at https://doi.org/10.5061/dryad.n02v6wx59[103].

Estimates of effective population size for the lean and siscowet morphs of *S. namaycush* derived from PSMC began at approximately 2.5 mya, similar to patterns among *Coregonus* species (Fig. 2, Supplementary Fig. 6a,c). From 2.5 mya to 200 kya both morphs showed a steady decline in $N_e$ with strong overlap between trajectories until reaching the lowest density of $N_e$ (~5,000) during the LGM. The $N_e$ curve using a generation time of 16 years (based on Hansen et al.[50]) and same salmonid mutation rate as employed for *Coregonus* above as in Crête-Lafrenière et al.[51] reaches its lowest point coincident with the LGM (Fig. 2, Supplementary Fig. 6). After the LGM, both *S. namaycush* morphs display a rapid expansion in $N_e$ subsequent to the expansion of *Coregonus* species, albeit with a short time lag (Fig. 2b).

Demographic analyses for the lean and siscowet morphs of *S. namaycush* with SMC++ showed consistent trajectories up to the LGM (Supplementary Fig. 6b,d, Supplementary Fig. 8). The estimates for both morphs began just before 2 mya and continued with a stepped decline to their lowest point, which is lower than any *Coregonus* species. Following this population decline, $N_e$ increased rapidly, coincident with the expansion of *Coregonus* populations. The lean morph of *S. namaycush* reached a maximum $N_e$ (~200,000) at 25 kya before steadily declining until the most recent estimate ($N_e$ ~ 50,000). Siscowet reached its maximum peak from 10–20 kya at ($N_e$ = ~150,000), and steadily declined until the most recent estimate ($N_e$ = ~50,000).

## Discussion

The *C. artedi* reference genome and 14 resequenced genomes from four species of Great Lakes *Coregonus* were used to explore population structure and species delimitation. Analyses of more than fifteen million polymorphic SNPs support differentiation of *C. artedi*, *C. hoyi*, *C. kiyi*, and *C. nigripinnis*, consistent with previous studies[23,52]. The two representatives of *C. nigripinnis* from Lake Nipigon were separated along the first PC from the rest of the Lake Superior populations, indicative of distinct population structure likely influenced by two drivers: (1) differing lineage contributions to populations in Lake Nipigon and Lake Superior, which occur at the contact zone of two glacial lineages of *Coregonus* spp. as inferred by Turgeon and Bernatchez[45] and Favé and Turgeon[53], and (2) drift from geographic

separation between Lake Nipigon and Lake Superior populations (Fig. 1b). Within Lake Superior, *C. artedi*, *C. hoyi*, and *C. kiyi* form distinct, non-overlapping clusters along PC2. Thus, variation in genome-wide SNPs among these species mirrors their differentiation in diet[19,23], spawning time and depth[54,55], and visual adaptation[24].

Genomic data provide a valuable resource for reconstructing effective population size through time and inform the timing of diversification. Utilizing two methods that rely on sequentially Markovian coalescent models, we estimated effective population size through time. The overall shape of the curves was similar across the two methods, and the timing of the lowest $N_e$ estimates were largely similar between PSMC and SMC++, with the latter method nadir coinciding with the start of the LGM (Supplementary Fig. 2, Supplementary Fig. 3). This result is not uncommon, as these methods have different accuracy at different time scales. SMC++ is considered more robust at recent time scales (up to 10 kya), while PSMC is more accurate at deeper time scales ( > 20 kya)[56,57]. A low point in $N_e$ for *Coregonus* species during the LGM suggests that suitable environmental conditions during this time were restricted while populations inhabited glacial refugia. Two glacial refugia have been proposed for *Coregonus* species, a Mississippian and Atlantic Coastal refugium, that ranged from lotic to offshore habitat[58]. Restrictions in habitat area would have constricted *Coregonus* population size while in these environments, reflected in the low estimates of $N_e$ found in PSMC and SMC++ analyses.

Our results imply that the four *Coregonus* species sampled had a shared $N_e$ trajectory back to ~80 kya, predating the LGM by nearly 50 kya. This shared trajectory of demographic history indicates that the *C. artedi* species complex was likely represented by a single ancestral species which diversified approximately 80 kya. We caution that estimates of divergence time through a visual examination of *Ne* trajectories could be confounded by other factors, such as post-divergence gene flow. Demographic independence prior to the LGM suggests that divergence occurred in the precursors to the Great Lakes, which were carved from multiple glacial cycles during the Pleistocene (past 2.4 million years)[59]. The timing of diversification and environmental context shown here contradicts previous hypotheses that divergence of the *C. artedi* species complex took place after the LGM[44–46,60,61]. The reconstructions of historical demography prompt two possible scenarios of divergence. In the first scenario, the common ancestor of the *C. artedi* complex diverged into independent, genetically distinct lineages with similar phenotypes and ecologies. These lineages were then isolated into the Atlantic and Mississippian glacial refugia during the LGM. Following the colonization of the Great Lakes after glacial retreat, secondary contact of glacial lineages drove speciation through niche specialization and character displacement as seen in ecomorphotypes of Lake Whitefish (*Coregonus clupeaformis*)[62,63]. A second plausible scenario is that the *C. artedi* complex progenitor specialized prior to the LGM due to the depth gradients present in the precursors to the Great Lakes. When confined to glacial refugia deepwater species, such as *C. hoyi* and *C. kiyi*, would not have been adapted to these shallower environments and would have had lower $N_e$ during that time as seen in Fig. 2. Additionally, natural selection on standing genetic variation in the form of rare-alleles conferring shallow-water phenotypes, and/or cross-species gene flow, could have played a role in minimizing habitat-phenotype mismatches during this time.

*Salvelinus namaycush* genome resequencing data provided a unique opportunity to investigate how a predator is affected by prey diversification and expansion. Analyses of genomic variation indicate strong genetic differentiation between lean and siscowet morphs of *S. namaycush* (Fig. 1c). These whole-genome patterns of variation reinforce differentiation of morphs previously discovered in mitochondrial[12], microsatellite[64], transcriptomic[65], and RAD-seq data[66].

Our demographic reconstructions show that *S. namaycush* effective population size was lower than *Coregonus* species during the LGM, but *S. namaycush* populations expanded rapidly following the population expansion and diversification of *Coregonus* species. The critical nature of *Coregonus* species as a primary food source for *S. namaycush*[67] plausibly signifies a causal, evolutionary response of predator diversification following

the expansion of prey into newly formed habitats following the LGM. Under a scenario of newly formed habitats, a predator would not likely expand into a new niche without a prey base to sustain it. More broadly, the demographic cycles observed in $N_e$ over evolutionary time are analogous (or perhaps even mechanistically 'homologous') to classic predator-prey population size oscillations over ecological time scales[68]. Similar to lags observed in Lotka-Volterra dynamics, we observed a slight lag in the expansion of *S. namaycush* populations following *Coregonus* population expansions. We hypothesize that *S. namaycush* diversified into morphs such as the siscowet to follow the newly-abundant deepwater *Coregonus* species (such as *C. kiyi* and *C. hoyi*) forage base, which is supported by contemporary diet studies[29–31].

Historical demography has been analyzed in both predator and prey radiations independently, yet co-evolutionary interactions of paired radiations inferred through coalescent approaches are rare. In one example, a study of a predator-prey-scavenger system used historical demography analysis of mitochondrial data to show that mammalian predators exhibited stable populations during prey population expansion[69]. Our results yielded different demographic patterns in the Great Lakes, where population size fluctuations are correlated across trophic levels. This may be inherently linked to the processes involved in radiation cascades. In the presence of diversifying prey, the predator species can evolve to (1) become generalists, or (2) become specialized to favor a distinct prey phenotype[7]. Thus, the second option, in tandem with habitat distinctiveness in prey, may lead to an upward adaptive radiation cascade through divergent selection in predator populations. In a parallel scenario, trophic interactions between the predator and prey prior to diversification could drive disruptive selection in the prey species, thereby promoting speciation in the prey complex[70]. However, diversification in prey would then provide the niche space for predator species to radiate. In either case, the diversification of the prey species is a prerequisite for predator radiation, as suggested here for native Great Lakes salmonid species. Taken together, this system clearly provides unique opportunities to explore the evolutionary patterns and processes of parallel radiations in evolutionary young species complexes.

Here we demonstrate that the North American *C. artedi* species complex is geologically young, yet older than previous hypothesized estimates, and that the species likely diverged around Marine Isotope Stage 5 (MIS5; 71 to 130 kya). MIS5 was characterized by warmer climate and minimal Laurentide Ice Sheet extent, similar to the Holocene, the current interglacial period[71,72]. The Great Lakes basins were formed by glacial erosion during at least two-dozen successive Pleistocene glaciations[59]. During MIS5, the Great Lakes likely had similar depths to the Holocene, as there was only one intervening glaciation that further eroded the basins. Thus, the Great Lakes during MIS5 likely had diverse lentic habitats that propelled *Coregonus* diversification into evolutionary distinct lineages, perhaps with accompanying phenotypic differentiation to match environmental gradients. Following the diversification of *Coregonus* around 80 kya, effective population size remained low or decreased. During this time, the climate was cool and the Laurentide Ice Sheet advanced and at times filled the Great Lakes basin with ice or silt-laden ice-sheet meltwater, setting an environmental stage where conditions for population expansion would be suboptimal.

The LGM confined *Coregonus* populations into isolated glacial refugia until the Laurentide Ice Sheet began to recede out of the Great Lakes basins around 17 kya[73]. Subsequent migration out of these refugial habitats into the newly reforming Great Lakes would have created scenarios of secondary contact between glacial lineages and abundant unoccupied niche space, conditions which can facilitate adaptive radiations[74–76]. In these vast deepwater environments, *Coregonus* progenitors were able to rapidly colonize and diversify into the remarkable diversity that was present prior to human-driven environmental degradation, population collapse, and loss of coregonine biodiversity during the 20th century.

The burst of rapid diversification within the *C. artedi* species complex occurred within the past 80 thousand years, equating to ~13,000 generations, a rate that would be ranked among the fastest in fishes estimated to-date (e.g., Hench et al.)[77]. This is also supported by previous studies that have

shown low levels of genetic divergence among these species[23,52,78]. The rapid diversification of the *C. artedi* complex is comparable to the speed of radiation and genetic divergence found in cichlid species in the African Rift Lakes[79]. Future studies should focus on the mechanistic drivers of this rapid diversification, such as further studies of genes under apparent natural selection that could shape the ecology of this group. Our findings substantiate the notion that the formation of deepwater habitats in the glacially eroded precursors to the Great lakes drove diversification in this species complex and following migration out of glacial refugia populations expanded rapidly.

The use of coalescent-based models with whole-genome data can help illuminate the timing and dynamics of diversification. We have shown that *Coregonus* species are likely a young adaptive radiation that underwent a rapid expansion in effective population size during MIS5 ( ~ 80 kya), a warm period when the Great Lakes had similar habitats to the Holocene. Moreover, we show that the subsequent population size expansion of multiple *Salvelinus namaycush* morphs closely follows the diversification of *Coregonus*, its primary prey species, suggesting linked eco-evolutionary dynamics. This work provides essential information to understand the timing and mechanisms of diversification in the commercially and ecologically important *Coregonus* and *Salvelinus* complexes. Finally, this study supports the hypothesis that habitat availability can promote diversification, which in turn suggests ongoing restoration efforts of Great Lakes ecosystems will yield important benefits for the conservation and management of native biodiversity.

## Methods

### *Coregonus artedi* genome: sample collection and DNA extraction

Eggs from *Coregonus artedi* were collected from Les Cheneaux Island, Lake Huron in 2015 and incubated in McDonald jars until hatching at the Great Lakes Science Center (Ann Arbor, MI). The fish were then moved to circular tanks, increasing in diameter from 1.22 to 3.05 m as the fish grew. Tank temperature was kept at 7–9 °C. A single mature female was euthanized with a lethal dose of buffered MS-222 and tissues (muscle, liver, eye, gut, and gills) were dissected immediately and preserved on dry ice. High molecular weight DNA was extracted from muscle using a Qiagen (Germantown, MD, USA) Genomic-tip 500/G kit and associated buffers following the manufacturer's protocol. Aliquots of extracted DNA were taken and Circulomics Short Read Eliminator (SRE) and SRE XL kits were used for size selection to preserve DNA fragments longer than 10 kb and 25 kb, respectively.

### Genome sequencing and assembly

Raw and size-selected genomic DNA was sequenced on fifteen MinION flow cells (version R9.4; Oxford Nanopore Technologies, Inc., Oxford, UK; hereafter ONT) using ligation sequencing kits (SQK-LSK-109). ONT long-reads were basecalled with high accuracy using Guppy v4.2.3+f90bd04. ONT reads with quality score <7 and/or length <1000 bp were discarded. Frozen muscle tissue was also sent to Novogene Corporation Inc. (Sacramento, CA, USA) for Illumina (San Diego, CA, USA) PE150 sequencing with the NovaSeq 6000 platform. Adapter trimming and quality filtering of the Illumina (San Diego, CA, USA) sequence data were performed with Trim Galore! v.0.6.0 (Krueger F. Trim-Galore!, http://www.bioinformatics.babraham.ac.uk/projects/trim_galore/) with a Phred quality score threshold of 20. To detect chromatin interactions an additional muscle sample was sent to Phase Genomics for Hi-C library preparation and Illumina (San Diego, CA, USA) Novaseq sequencing.

An initial genome assembly was produced with Flye v2.8-b1674[80] using the genome size parameter (-g) set to 2.6 Gb and the maximum overlap distance "-m" set to 10 kb. Two iterations of internal polishing with Flye were run using the iteration parameter ("-i 2"). Following the initial assembly Illumina short reads were utilized for polishing. Illumina short reads were mapped to the initial genome assembly with bwa v0.7.17-r1198[81] to create a sequence alignment, which was passed to Pilon v1.23[82] for two rounds of polishing. Purge Haplotigs v1.1.2[83] was used to identify and

remove haplotigs and deep coverage repeats. Kraken2 v.2.0.8-beta[84] was used to identify and remove possible contamination (e.g., microbiome sequences) using a custom database comprised of the Atlantic salmon (*Salmo salar*) genome "ICSASG_v2"[85] (GCF_000233375.1) and sequences from archaea, bacteria, plasmid, viral, human, fungi, plant, protozoa and the "nr" and "nt" databases from NCBI.

We took a two-step approach to scaffold the draft assembly. First, we used Hi-C to identify well-defined blocks of three-dimensional structure, which were then integrated into an existing *C. artedi* linkage map. A Hi-C contact heatmap was created with Juicer v1.6[86]. Using the aligned Hi-C reads, misjoins were corrected with 3D-DNA v18092[87] with the settings "--editor-coarse-resolution 100000 -i 50000 -q 20 --editor-coarse-stringency 20". Contact maps were visualized with Juicebox assembly tools[88], and the remaining misassemblies were manually corrected. The output was passed to an additional run of 3D-DNA to finalize the manual edits, and place gaps between scaffolds filled with 150 N's. The Hi-C scaffolded assembly was then integrated with the female *C. artedi* linkage map from Blumstein et al.[47] using Chromonomer v1.13[89]. A file containing contigs and gaps was generated with a custom python script 'fasta_to_simple_agp.py'. The paired-end markers from the linkage map were then mapped to the Flye assembly version with bwa v 0.7.17-r1198[81]. The linkage map markers, assembly gap file, and sequence alignment file were used as inputs for Chromonomer v1.13[89] to integrate Flye contigs into the linkage map, resulting in a merged Hi-C and linkage map-scaffolded assembly. Genome completeness was assessed for both the initial Flye assembly and final assembly with BUSCO v5.1.2[90] using the actinopterygii_odb10 database.

Repetitive elements in the assembly were first characterized with RepeatModeler v2.0.1[91] with the "-LTRStruct" flag enabled. The repeat library was checked for coding sequences with blastx v2.9.0+ [92] against the uniport_sprot database[93], which was filtered for transposable elements. If there was a protein match, those sequences were removed from the repeat library. The protein-filtered repeat library was fed into RepeatMasker v4.1.1[94], which was run two ways. First, a gff file was generated from predicted repeats, then another pass that incorporated the "vertebrata" database from Dfam[95]. The results from each pass were combined and produced a final gff containing locational information for repetitive sequences.

### *Coregonus* species sampling and genome resequencing

Samples of *C. artedi*, *C. hoyi*, and *C. kiyi* were collected from multiple sites in Lake Superior between May and August of 2015 and *C. nigripinnis* from Lake Nipigon in September 2018 (Supplementary Table 1). For the Lake Superior fish, multiple gear types were used for sample collection across a depth gradient with a 15.24 m midwater trawl used for the *C. artedi* samples, and a 11.89 m benthic trawl for the *C. hoyi* and *C. kiyi* samples. The *C. nigripinnis* samples were collected with multi-panel gill nets of various mesh sizes. Samples were identified to species following Koelz[54] and Eshenroder et al.[15], euthanized by pithing, and dissected to obtain muscle tissues for DNA extraction with Purelink Genomic DNA Kit (Invitrogen, Inc; Carlsbad, CA, USA). Four individuals from each species were extracted except *C. nigripinnis* (*n* = 2). DNA was sent for Illumina PE150 genomic resequencing on the NovaSeq 6000 platform performed by Novogene (Davis, CA, USA). Illumina-specific sequencing adapters were trimmed from the reads using TrimGalore! v0.6.6 (Krueger F. Trim-Galore!, accessible at http://www.bioinformatics.babraham.ac.uk/projects/trim_galore/) implementing a Phred quality score cutoff of thirty (-q 30) and default error rate (-e 0.1).

### Population genomic structure

Genome resequencing data for individuals of each species of *Coregonus* were mapped to the genome assembly using bwa 'mem' v0.7.17[81]. Variant calling and filtering across individual samples were completed with BCFtools v1.15[96] 'mpileup' (-C50) and 'call' using default parameters. VCFtools v0.1.17[97] was used to filter only bi-allelic SNPs with minor allele frequency > 0.05 and with no missing data. Visualization of population genomic structure was completed in R v4.3.0 by first converting the VCF to genlight object with vcfR v1.10.0, which was followed by principal

component analysis (PCA) completed with the 'glPca' function from adegenet v.2.1.3[98]. In addition, true singletons were removed using VCFtools and maximum likelihood phylogenetic analyses were performed using IQ-Tree v2.2.2.6[99]. The optimal substitution model was selected using ModelFinder v2.2.2.6[100], only allowing models that included an ascertainment bias correction since the alignment contained only variable sites. Branch support was assessed using 1,000 ultrafast bootstrap replicates[101]. Additional optimization of the bootstrap trees using nearest neighbor interchange (-bnni) was performed to minimize the potential impacts of model violations due to concatenating a genome-wide markers. The resulting unrooted phylogeny was visualized using FigTree v.1.4.4. (http://tree.bio.ed.ac.uk/software/figtree/).

### *Coregonus* demographic analysis with PSMC

For demographic history analyses, the resequenced *Coregonus* spp. were mapped to the repeat masked version of the genome and variants were called in the same manner as the population structure analyses. PSMC requires consensus genome fastq files for each sample, which were produced with the vcfutils.pl v0.1.17[97] vcf2fq command with a minimum depth of a third of the average coverage and a maximum depth of two times the average coverage. PSMC v0.6.5-r67[34] was run for 25 iterations using the parameters -t5 -r5 -p "$4 + 20*2 + 6*4 + 4$" with $t$ being the upper limit of time for the most recent common ancestor, $r$ being a ratio of mutation rate over the recombination rate ($\theta$/p), and $p$, which is the number of free atomic time intervals. PSMC analysis was assessed for variance via bootstrapping with 100 replicates with replacement by splitting the consensus genome sequence for each individual into five Mb blocks and estimating $N_e$.

Plotting of PSMC output requires estimates for mutation rate and generation time. A nuclear genome mutation rate of $7.26 \text{ e}^{-09}$ mutations/site/generation was selected following Crête-Lafrenière et al.[51], a rate that falls within the confidence interval for *S. salar* presented by Bergeron et al.[48]. PSMC plots were also generated with the lower ($2.50 \text{ e}^{-09}$) and upper ($8.23 \text{ e}^{-09}$) bounds of the confidence interval from Bergeron et al.[48] to evaluate the effects that mutation rate had on the demographic analyses. Generation time was calculated by using age-specific elements in a life table (Appendix I) and a final generation time of six years was applied to all samples. The parameters were input into the PSMC utility 'psmc_plot.pl' v0.6.5-r67 and applied to the combined PSMC bootstrap results for each sample with the '-R' flag enabled to output the data as text files. The final output data were plotted with a custom R script.

### *Coregonus* demographic analysis with SMC++

To examine cisco demographic histories at more recent time-scales we employed SMC++ v1.15.2[35]. Unlike PSMC, SMC++ requires multiple sampled genomes per population or species[35]. VCF files from mapping to the repeat masked genome were merged for each species using BCFTools v.0.1.19[96], then filtered to retain only biallelic SNPs (-m2 -M2 -v snps). The merged VCF files were converted to SMC++ input format using the 'vcf2smc' subcommand. This command was used separately for each of the 38 *Coregonus* linkage groups and for every possible "distinguished lineage" within each species. Demographic histories for each species were then reconstructed using the 'estimate' subcommand, with 50 EM iteration, polarization error set to 0.5 (indicating an unknown ancestral allele), and 40 spline knots. The minimum model timepoint was set to 10 generations and the maximum model time point was set to 10,000 generations. However, we note that specification of the maximum model time point does not appear to work as intended; instead, the software seems to use the heuristic approach to automatically calculate model time points. As with PSMC, we used a series of mutation rates to evaluate their impacts on $N_e$ estimates, with $7.26 \text{e}^{-09}$ mutations/site/generation used as the final rate, with the most support[51]. To assess confidence in the inferred demographic history we performed 100 non-parametric bootstrap replicates using a custom script to resample the original VCF files in 5 Mb blocks. For each bootstrap replicate, SMC++ was run using the parameters described above. Results were plotted using a custom R script with a generation time of 6 years.

### *Salvelinus namaycush* population structure and demographic analysis

Historical demographic analysis of *S. namaycush* generally followed the *Coregonus* analysis. *S. namaycush* from Lake Superior comprised of two different morphs were sequenced to ≥15x coverage with Illumina PE150 genomic sequencing on the NovaSeq 6000 platform performed by Novogene (Davis, CA, USA). The samples were comprised of three lean and three siscowet *S. namaycush* individuals. *S. namaycush* sequence data were processed with the same trimming and quality filtering steps as the *Coregonus* data. Reads were mapped to a repeat-masked version of the *S. namaycush* reference genome, "SaNama_1.0"[49] using bwa v0.7.17[81] and SNP calls were conducted with BCFtools v.0.1.19[96]. Population structure analyses were completed with the same techniques as *Coregonus* species. Demographic history estimation was then completed using PSMC and SMC++ as per *Coregonus* species analyses, except for the generation time. *S. namaycush* was assigned a generation time of 16 years based on previous work that showed they reach sexual maturity exceeding 15 years in Canadian lakes[50]. A series of generation times (6, 8, 10, 12, 14, 16, 18, 20 years) were also applied when plotting the results of PSMC to estimate the effect of differential generation times on trajectories of $N_e$.

### Reporting summary

Further information on research design is available in the Nature Portfolio Reporting Summary linked to this article.

## Data availability

The genome assembly and all sequence data are available under NCBI BioProjects PRJNA1062807 (*Coregonus* spp.) and PRJNA1077361 (*Salvelinus namaycush*). Source data for the figures can be found at https://doi.org/10.5061/dryad.n02v6wx59.

## Code availability

Code for data analysis is available at https://github.com/KrabbenhoftLab/Coregonus_demography.

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

## Acknowledgements

We thank the U.S. Geological Survey Research Vessel *Kiyi* Captain Joe Walters, First Mate Keith Peterson, and Engineer Charles Carrier. We would also like to thank the Red Cliff Band of the Lake Superior Chippewa Nation and members of OMNDMNRF (Anders Nyman, David Niskanen, and Robyn Avis, Eric Berglund and David Montgomery) for providing samples. This study was supported by the Great Lakes Fishery Commission (Awards #2018-KRA-44073 and #2021-KRA-440960 to T.J.K) and the University at Buffalo College of Arts and Sciences (Dean's Fellowship awarded to N.J.C.B.). Any use of trade, product, or firm names is for descriptive purposes only and does not imply endorsement by the U.S. Government. All sampling and handling of fish were carried out in accordance with guidelines for the care and use of fishes by the American Fisheries Society (Jenkins et al., 2014). We would also like to thank Joseph R. Tomelleri for giving permission to use his artwork in the manuscript.

## Author contributions

T.J.K., N.J.C.B., D.L.Y., A.S.A., W.S., V.A.A., and L.B. conceived the study. D.L.Y. and W.S. provided samples. N.J.C.B., D.J.M., C.A.O., M.A.B., D.L.Y., E.N., and T.J.K. generated data and conducted analyses. N.J.C.B. and E.N. assembled and scaffolded the *Coregonus artedi* genome. E.K.T. provided context and interpretation of geological data. N.J.C.B. and T.J.K. wrote the initial draft, and all co-authors edited the manuscript.

## Competing interests

The authors declare no competing interests.
