## [Peer Review File · Communications Biology]

Reviewers' comments:

Reviewer #1 (Remarks to the Author):

The authors performed genome sequencing and/or resequencing of four *Coregonus* species and their predator *Salvelinus namaycush* to examine the origin of Laurentian Great Lakes fish fauna through upward adaptative radiation cascade. Generally speaking, the overall writing of this interesting topic is very good, and numerous data were generated to support the main conclusions. However, minor revisions are still required before acceptance for publication.

1. The format of reference citation in the main text should be consistent with the journal guidance.
2. Lines 149-150: What is the size for the Nanopore reads?
3. Lines 153-155: Why didn't you perform Hi-C sequencing to construct "real chromosomes"? That is to say, the reference genome assembly can be improved with additional Hi-C data. That is why complete BUSCOs are relatively low (line 156).
4. Lines 167-169: Except for *C. artedi*, the other three *Coregonus* species were sequenced with only short-read Illumina sequencing. In this case, the comparative results may be unbelievable. Provide more discussions about this issue.
5. In the Methods section, provide more details (such as company, city, and country names, or any necessary reference) about important reagents, devices, and software.
6. Lines 477-480: Genomic data should be released as soon as possible for public availability.

Reviewer #2 (Remarks to the Author):

The manuscript first describes the establishment of a reference genome assembly for *Coregonus artedi* based on Nanopore (28x) and Illumina (40x) sequencing data. Compared to genome assemblies of other species within *Coregonus*, the genome assembly remains a bit more fragmented and has a lower BUSCO score. The manuscript then further makes use of resequencing data for 4 *Coregonus* species (14 individuals) and 2 *Salvelinus namaycush* morphs (6 individuals) that all co-occur in the Great Lakes to infer the demographic histories of these six species. As both lineages have diverged along depth, the manuscript test if the divergence times of the species is consistent with an upward adaptative radiation cascade.

The manuscript is very well written, easy to follow, and empirically testing the conceptual idea of an upward adaptative radiation cascade is novel and very interesting.

However, I feel the results are presented too affirmative given various limitations of the results and inference approaches used. Specifically, I have some concerns regarding the quality of the reference genome assembly, and the interpretation of the divergence time inferences.

1) Quality of reference genome assembly:

The manuscript lacks a critical assessment of the quality of the produced reference genome. A more thorough comparison with reference genome assemblies for other *Coregonus* and salmonid

species would be warranted. Potentially even an evaluation to which respect and for which analysis it is preferred to use a phylogenetically closer references versus a more continuous, more complete reference of a more distantly related species.

One major concern I see with the assembly is the discrepancy between the estimated genome size of 3.18 Gb and the assembly size of 2.06 Gb. It might be worth to run a kmer analysis using the Illumina data to get another genome size estimate. Further this discrepancy likely suggests that duplicated regions have been collapsed. To some extent this has been seen also in other salmonid assemblies, but it would be worth to characterize this in more detail. Identifying such collapsed regions enables their exclusion or masking in downstream analysis (e.g., in demographic inference which might likely be affected by it).

In addition, the resequencing data mapped to the assembly results in a somehow low breadth of coverage, which might indicate additional issues with the assembly. Any idea why this is happening?

1) Interpretation of divergence time inference:

The manuscript misses to interpret the PSMC-based analysis within the context of its limitations, such as uncertainty in generation time and mutation rate. While the manuscript compiles life tables to infer generation times for the different species, and hence aims to minimize uncertainty regarding generation times, the mutation rate used still comes with a big uncertainty. Uncertainty regarding the mutation rate might affect the interpretation suggested in the manuscript that *Coregonus* radiated before LGM. This will need to be discussed in the manuscript and taken into account when interpreting the results.

And not only a difference in mutation rate might shift the N_e curve along the x-axis. The inference of when the *Coregonus* species diverged from each other could have also been influenced by post divergence gene flow, likely to have occurred in a scenario of divergence in sympatry. Further, as already pointed out above, I would expect that the duplicated and collapsed regions of the genome influence heterozygosity calls and hence the demographic inference. Further given the choice of using -D 120 which deviates from the recommendations (“It is recommended to set -d to a third of the average depth and -D to twice.”), I am concerned that the demographic inference is indeed influenced by such issues. On a similar note, the manuscript is not clear regarding the masking used for running SMC++. Was a BED-formatted mask file used marking missing data and delineating large uncalled regions and regions with poor mapping quality. For both demographic inference approaches it might be worth considering to identify regions in the assembly with poor mappability using a k-mer approach (<https://lh3lh3.users.sourceforge.net/snpsable.shtml>) and exclude those as well.

While my above comment concern absolute time estimates for a comparison with the LGM, there are other (difference in reference genome quality, depth of coverage, difference in generation time)

that might affect the interpretation regarding the earlier split between the *Coregonus* species relative to the *Salvelinus* species. I am not sure the difference in reference genome quality affects the demographic inference but maybe the authors could comment on it. However, it has been demonstrated previously that low genomic coverage shifts the PSMC trajectory (for example Nadachowska-Brzyska et al. *Molecular Ecology* 2016 doi: 10.1111/mec.13540). To which respects it affects the analysis here could be explored by down sampling to a common coverage and/or omitted by providing heterozygosity estimates for all samples (could be added to S1 and S2). Regarding the difference in generation time between the species I appreciate the efforts presented in the Supplement, but I am still wondering how plots with both species would look like if the same generation time is used for both species.

I think the mismatch between the two methods deserves more attention and needs to be somehow reconciled. As presented right now would the two approaches not suggest different biological interpretations?

Minor comments:

Line 123 referencing Cahill et al. for the approach used in this study feels a bit misleading, as the study actually suggests a different approach (hPSMC), which is also used in some of the other references listed (Foote et al., Prüfer et al, Van der Valk et al.)

This also made me wonder to which respect the hPSMC approach might provide additional insights. And to which respect using SCM++ with two population input to infer divergence time might be of interest.

SMC++ is optimized for hundreds of samples and if used otherwise requires finding the optimal settings. How was that achieved and can the reader get a sense of whether the optimizer is well fitted and regulated.

Busco results as plotted in Figure S1, might be better suited to be presented in a table.

Why are there 3 *Salvelinus* lines in Figure 8b? Shouldn't it be two, one for each of the morph?

Philine Feulner

Reviewer #3 (Remarks to the Author):

1. The paper aims is to show the radiation of species of the genus *Coregonus* as well as two morphs of its predatory species, *Salvelinus namaycush*, by using demographic predictions back in time by means of two coalescent-based methods. A seemingly more secondary objective was to describe the population structure of the two complexes.

2. The paper is well written and easy to read and accomplishes what it is promised in the intro. I find that looking into the demographic story of closely related species is a clever and useful way in which species radiation can be investigated. In turn, a good knowledge of the environmental history where the species are present can hint to the conditions that promoted the radiation in the first place. Additionally, and not less important, the authors have generated a full genome assembly of one of the species, which in itself is a valuable resource that researchers can use in further work. Thus, I do believe that this work is of great interest for speciation and evolutionary research in general.

3. The paper settles upon previous research on the species and elaborates henceforth, making its conclusions convincing. Adding to the novelty of the research, it shows its conclusions to contrast with previous understanding, indicating that radiation likely occurred before and not after LGM. I find very sharp the authors consider a first scenario of "neutral" diversification first and thereafter a selective radiation and a second where selective radiation is the origin of species diversification.

4. Concerning lines 52-53, I have inspected this article to check for possible more recent literature than Johnson et al., 1996 -> <https://doi.org/10.1093/jhered/esz075> and am now not so convinced that this claim the authors make that cases are infrequent is valid. Please, consider this.

5. Upon some research on the methods used, it seems that coalescent methods need to have a good account of the assumptions needed for the different models. However, I did not find statements that clarify this. Please, consider.

6. Having more experience on population genetics myself, my main concerns relate to the results shown in the PCA. For a more fully understandable genetic structure, I find that it is required to separate neutral from non-neutral markers. I understand the authors not making this distinction as this was not the main objective of the study (and the amount of methods required increases, although perhaps an exploration using only BAYESCAN, or PCAdapt or any other selection-scanning software can be done without spending much time).

7. Sex of the individuals have not been reported, neither any signal that shows which markers are sexual (or if there is sexual chromosomes). Note that population structure results can change if not considering sex, and in this case, where the sample size is not big could bias the results substantially. For more information, see this DOI: 10.1111/mec.14217 and 10.1111/eva.12979.

8. I have also a minor concern concerning the sampling scheme. Since the sample size is not big enough the choice of sampling coordinates may bias population structure results. However, I realise that given the low genetic differentiation between species, maybe this is a least concern.

Reviewers' comments:

We would like to thank the reviewers for the helpful and constructive comments. The input provided has significantly improved the quality of the manuscript, especially through the reanalysis of demographic data and new genome assembly suggested by the reviewers.

Reviewer #1 (Remarks to the Author):

The authors performed genome sequencing and/or resequencing of four *Coregonus* species and their predator *Salvelinus namaycush* to examine the origin of Laurentian Great Lakes fish fauna through upward adaptative radiation cascade. Generally speaking, the overall writing of this interesting topic is very good, and numerous data were generated to support the main conclusions. However, minor revisions are still required before acceptance for publication.

1. The format of reference citation in the main text should be consistent with the journal guidance.

The reference citation format has been updated to fit the submission style guide for Springer Nature.

2. Lines 149-150: What is the size for the Nanopore reads?

Basecalled nanopore reads ranged in length from 1 to 336,024 bp. For the assembly, reads less than 1000 bp were discarded. Additionally, using the assembly software Flye, parameters were set that did not incorporate many of the smaller reads into the assembly. We inserted the following text: "ONT reads with quality score <7 and/or length <1000 bp were discarded (lines 382-383)."

3. Lines 153-155: Why didn't you perform Hi-C sequencing to construct "real chromosomes"? That is to say, the reference genome assembly can be improved with additional Hi-C data. That is why complete BUSCOs are relatively low (line 156).

Thank you for pointing this out, we have included an updated *C. artedi* assembly to satisfy additional concerns presented by reviewer two. This updated assembly incorporates results of both a linkage map and Hi-C sequencing to construct our scaffolds, which now reflect the chromosomes (lines 154-155). Complete BUSCOs for the new genome assembly increased by ~10% when compared to the previous assembly. Assembly contiguity was also greatly improved (scaffold N50 62.61 mb).

4. Lines 167-169: Except for *C. artedi*, the other three *Coregonus* species were sequenced with only short-read Illumina sequencing. In this case, the comparative results may be unbelievable. Provide more discussions about this issue.

A combination of Oxford Nanopore long-reads and Illumina short-read data were utilized to assemble a representative of *C. artedi*. All samples (including *C. artedi*) in the demographic history estimation were from lakes Superior or Nipigon and were resequenced with only paired-end Illumina, which we indicate in lines 481-482. Thus, our analyses (population structure, demographic history) all utilized only short-read Illumina data, while the reference assembly was based on nanopore reads.

5. In the Methods section, provide more details (such as company, city, and country names, or any necessary reference) about important reagents, devices, and software.

Change made. Software has been cited under the guidelines of the style guide of Communications Biology.

6. Lines 477-480: Genomic data should be released as soon as possible for public availability.

The updated version of the genome assembly and raw data will be released on NCBI upon acceptance of this manuscript.

Reviewer #2 (Remarks to the Author):

The manuscript first describes the establishment of a reference genome assembly for *Coregonus artedi* based on Nanopore (28x) and Illumina (40x) sequencing data. Compared to genome assemblies of other species within *Coregonus*, the genome assembly remains a bit more fragmented and has a lower BUSCO score. The manuscript then further makes use of resequencing data for 4 *Coregonus* species (14 individuals) and 2 *Salvelinus namaycush* morphs (6 individuals) that all co-occur in the Great Lakes to infer the demographic histories of these six species. As both lineages have diverged along depth, the manuscript tests if the divergence times of the species is consistent with an upward adaptive radiation cascade.

The manuscript is very well written, easy to follow, and empirically testing the conceptual idea of an upward adaptive radiation cascade is novel and very interesting.

However, I feel the results are presented too affirmative given various limitations of the results and inference approaches used. Specifically, I have some concerns regarding the quality of the reference genome assembly, and the interpretation of the divergence time inferences.

1) Quality of reference genome assembly:

The manuscript lacks a critical assessment of the quality of the produced reference genome. A more thorough comparison with reference genome assemblies for other *Coregonus* and salmonid species would be warranted. Potentially even an evaluation to which respect and for which analysis it is preferred to use a phylogenetically closer reference versus a more continuous, more complete reference of a more distantly related species.

One major concern I see with the assembly is the discrepancy between the estimated genome size of 3.18 Gb and the assembly size of 2.06 Gb. It might be worth to run a kmer analysis using the Illumina data to get another genome size estimate. Further this discrepancy likely suggests that duplicated regions have been collapsed. To some extent this has been seen also in other salmonid assemblies, but it would be worth to characterize this in more detail. Identifying such collapsed regions enables their exclusion or masking in downstream analysis (e.g., in demographic inference which might likely be affected by it).

The new version of the genome assembly (2.491 Gb) is closer to the estimates provided by Feulgen densitometry. It is still possible that regions in the new assembly are collapsed, however the current assembly size is within the range of other *Coregonus* genome assemblies available such as *C. lavaretus* balchen [2.068 Gb; De-Kayne et al. (2020)] and *C. clupeaformis* [2.628 Gb; Merot et al. (2022)]. Please see Table 1; only 137 Kb of our assembly is not placed into one of the 38 chromosomes. Additionally, Pflug et al. (2020) (<https://doi.org/10.1534/g3.120.401028>) has shown that kmer based estimation of genome size can be 4.5% less than actual genome size.

In addition, the resequencing data mapped to the assembly results in a somehow low breadth of coverage, which might indicate additional issues with the assembly. Any idea why this is happening?

The breadth of coverage using the previous assembly was low (~80%) due to overall genome quality, and only running analyses on the 38 linkage groups. With the new version of the genome these values have

increased to ~99% (Supplementary Table 2).

1) Interpretation of divergence time inference:

The manuscript misses to interpret the PSMC-based analysis within the context of its limitations, such as uncertainty in generation time and mutation rate. While the manuscript compiles life tables to infer generation times for the different species, and hence aims to minimize uncertainty regarding generation times, the mutation rate used still comes with a big uncertainty. Uncertainty regarding the mutation rate might affect the interpretation suggested in the manuscript that *Coregonus* radiated before LGM. This will need to be discussed in the manuscript and taken into account when interpreting the results.

We agree that methods like PSMC and SMC++ have inherent limitations. We aimed to minimize possible biases in three ways: (1) inclusion of multiple methods for inferring demographic history (PSMC and SMC++), (2) including bootstrap replicates to assess variation in N_e estimates, (3) addition of mutation rate analyses (see Supplement for new details). Briefly, three mutation rates were tested to understand the sensitivity of the PSMC and SMC++ analyses to differences in mutation rate. We chose to use the lower ($2.50e^{-09}$) and upper ($8.23e^{-09}$) bounds of the 95% confidence interval of mutation rate estimates from Bergeron et al. (2023), in addition to our original mutation rate ($7.26e^{-09}$) derived from Crête-Lafrenière et al. (2012). Please see discussion of these methods on lines 514-519 & 537-540. The results are found in Supplementary Figure 4. Two of the mutation rates ($7.26e^{-09}$ and $8.23e^{-9}$) produced congruent results for both PSMC and SMC++ analyses. PSMC analyses using the third (and lowest) mutation rate ($2.50e^{-09}$) shifted the demographic history curve back in time along the x-axis. Interestingly, these shifted PSMC demographic curves more closely match the SMC++ demographic curves (e.g. N_e nadir just prior to the LGM). For SMC++ analyses, the lowest mutation rate did not shift the demographic history curve, although the analyses identify additional peaks in N_e that occurred from 30 – 500ka. Two of the four species (*C. artedi* and *C. hoyi*) still exhibit N_e nadirs immediately prior to or during the LGM. The other two species (*C. kiyi* and *C. nigripinnis*) exhibit declines in N_e near the LGM, but these declines are smaller in magnitude than earlier crashes. Therefore, although choice of mutation rate does affect the estimated timing and trajectories of the N_e curves, we believe that our interpretation of the demographic histories of these species best fits the data available and largely consistent across methods.

And not only a difference in mutation rate might shift the N_e curve along the x-axis. The inference of when the *Coregonus* species diverged from each other could have also been influenced by post divergence gene flow, likely to have occurred in a scenario of divergence in sympatry. Further, as already pointed out above, I would expect that the duplicated and collapsed regions of the genome influence heterozygosity calls and hence the demographic inference. Further given the choice of using -D 120 which divides from the recommendations (“It is recommended to set -d to a third of the average depth and -D to twice.”), I am concerned that the demographic inference is indeed influenced by such issues. On a similar note, the manuscript is not clear regarding the masking used for running SMC++. Was a BED-formatted mask file used marking missing data and delineating large uncalled regions and regions with poor mapping quality. For both demographic inference approaches it might be worth considering to identify regions in the assembly with poor mapability using a k-mer approach (<https://lh3lh3.users.sourceforge.net/snpable.shtml>) and exclude those as well.

When extracting variants from the VCF file for each of the samples we filtered based on read depth as described above (i.e., (“It is recommended to set $-d$ to a third of the average depth and $-D$ to twice.”) (lines 502-504). Additionally, in our revised manuscript, we took steps to identify and mask repetitive elements in the genome prior to demographic analysis. Repetitive elements comprise ~70% of the genome and would have contributed to poor read mapping. Ultimately, the demographic history trajectories based on unmasked versus masked genomes were highly similar in shape, but with overall lower N_e estimates through time in the masked version. Overall, the masking of repetitive elements in the reference genome reduced the number of SNPs called at repetitive loci increasing the accuracy of N_e estimates. We thank the reviewer for this helpful suggestion!

While my above comment concern absolute time estimates for a comparison with the LGM, there are other (difference in reference genome quality, depth of coverage, difference in generation time) that might affect the interpretation regarding the earlier split between the *Coregonus* species relative to the *Salvelinus* species. I am not sure the difference in reference genome quality affects the demographic inference but maybe the authors could comment on it. However, it has been demonstrated previously that low genomic coverage shifts the PSMC trajectory (for example Nadachowska-Brzyska et al. Molecular Ecology 2016 doi: 10.1111/mec.13540). To which respects it affects the analysis here could be explored by down sampling to a common coverage and/or omitted by providing heterozygosity estimates for all samples (could be added to S1 and S2). Regarding the difference in generation time between the species I appreciate the efforts presented in the Supplement, but I am still wondering how plots with both species would look like if the same generation time is used for both species.

Please see our above responses regarding genome assembly quality and coverage of the resequencing data.

Supplementary Figure 8 shows the N_e estimates from PSMC for *S. namaycush* if we set the generation time to the same value as the *Coregonus* spp. The curve would be shifted well forward of the current state and would further indicate that there is a time lag between the expansion of *Coregonus* spp. and *S. namaycush*. With a generation time of 6 years, the *S. namaycush* samples would also have a high N_e during the LGM. This is a biologically unrealistic scenario, as these species ranges and available habitats would have been confined to glacial refugia. A generation time of six years for *Salvelinus* is inconsistent with contemporary life history data and yields demographic history results that would be difficult to understand in light of geologic history.

I think the mismatch between the two methods deserves more attention and needs to be somehow reconciled. As presented right now would the two approaches not suggest different biological interpretations?

We agree with the reviewer about this issue. Our re-analyses using masked genomes (both species), more appropriate read mapping parameters (both species), and a significantly higher-quality assembly (*Coregonus artedii*) resulted in much closer agreement between PSMC and SMC++ demographic results. For the *Coregonus* species, the N_e nadir from PSMC occurs at the end or just after the LGM [19-33 kya], while the N_e nadir from SMC++ occurs at the start or just prior to the LGM [19-33 kya]. We observed a

similar improvement for *S. namaycush*; the PSMC nadir occurs at the end of the LGM [17-25 kya], while the SMC++ nadir occurs just prior to the LGM [50-60 kya].

Yes, improvements to the analysis yielded much higher agreement between methods, now with only slightly different stories about the demographic history of these species. However, both analyses point toward the LGM being a pivotal point in the demographic histories of all analyzed species. All species experienced dramatic N_e declines leading up to the LGM, followed by massive and rapid increases in N_e . Additionally, the demographic histories of the two radiations diverged well before the LGM in both sets of analyses. Improvements to the genome assembly significantly reduced this mismatch between methods, (hopefully) mitigating the reviewer's concern.

Minor comments:

Line 123 referencing Cahill et al. for the approach used in this study feels a bit misleading, as the study actually suggests a different approach (hPSMC), which is also used in some of the other references listed (Foote et al., Prüfer et al, Van der Valk et al.)

This also made me wonder to which respect the hPSMC approach might provide additional insights. And to which respect using SMC++ with two population input to infer divergence time might be of interest.

Cahill et al. (2016), indicate that hPSMC is highly sensitive to post-divergence gene flow, and is powerful in inferring the timing of the end of gene flow between diverging populations. Studies such as Bernal et al (2022) indicate differentiation between *C. artedi* and *C. kiyi* however other *Coregonus* spp. in Lake Superior lack differentiation, therefore applying hPSMC to this system could lead to an unreliable estimate of divergence time for this clade. We attempted to run the SMC++ "split" command to infer divergence times. Unfortunately, we found that this function does not appear to work as intended. Others have pointed out similar issues (e.g. <https://github.com/popgenmethods/smcpp/issues/243>, <https://github.com/popgenmethods/smcpp/issues/195>, <https://github.com/popgenmethods/smcpp/issues/189>).

SMC++ is optimized for hundreds of samples and if used otherwise requires finding the optimal settings. How was that achieved and can the reader get a sense of whether the optimizer is well fitted and regulated.

The EM algorithm used by SMC++ always successfully converged using the default optimization criteria. Additionally, the demographic histories estimated by SMC++ were generally consistent with the results of PSMC (especially after inclusion of the new genome version), suggesting that the SMC++ settings we used were appropriate.

Busco results as plotted in Figure S1, might be better suited to be presented in a table.

Done. We have removed the Supplementary Figure 1 and reported the genome assembly statistics in an updated version of Table 1.

Why are there 3 Salvelinus lines in Figure 8b? Shouldn't it be two, one for each of the morph?

We initially uploaded an incorrect version of the figure; we thank the reviewer for catching our mistake!
The figure has also now been updated with the newest analysis as well.

Philine Feulner

Reviewer #3 (Remarks to the Author):

1. The paper aims to show the radiation of species of the genus *Coregonus* as well as two morphs of its predatory species, *Salvelinus namaycush*, by using demographic predictions back in time by means of two coalescent-based methods. A seemingly more secondary objective was to describe the population structure of the two complexes.

2. The paper is well written and easy to read and accomplishes what it is promised in the intro. I find that looking into the demographic story of closely related species is a clever and useful way in which species radiation can be investigated. In turn, a good knowledge of the environmental history where the species are present can hint to the conditions that promoted the radiation in the first place. Additionally, and not less important, the authors have generated a full genome assembly of one of the species, which in itself is a valuable resource that researchers can use in further work. Thus, I do believe that this work is of great interest for speciation and evolutionary research in general.

3. The paper settles upon previous research on the species and elaborates henceforth, making its conclusions convincing. Adding to the novelty of the research, it shows its conclusions to contrast with previous understanding, indicating that radiation likely occurred before and not after LGM. I find very sharp the authors consider a first scenario of "neutral" diversification first and thereafter a selective radiation and a second where selective radiation is the origin of species diversification.

We appreciate the positive comments from the reviewer and agree about the broad interest of the system and our findings for speciation and evolutionary research.

4. Concerning lines 52-53, I have inspected this article to check for possible more recent literature than Johnson et al., 1996 -> <https://doi.org/10.1093/jhered/esz075> and am now not so convinced that this claim the authors make that cases are infrequent is valid. Please, consider this.

Thank you for bringing this point to our attention. We have changed the wording to remove the premise of this phenomenon infrequently occurring. This is valid given that other model systems in radiations such as African Rift Lake cichlids also have sympatric lineages that underwent adaptive radiations.

5. Upon some research on the methods used, it seems that coalescent methods need to have a good account of the assumptions needed for the different models. However, I did not find statements that clarify this. Please, consider.

The two primary determinants on the output of PSMC and SMC++ are assumptions of mutation rate and generation time. We have addressed additional comments by reviewer 2 above regarding these assumptions. In the revised version of the manuscript we have conducted sensitivity analyses of different mutation rates. Please see the methods outlined on lines 514-519 and 537-540.

6. Having more experience on population genetics myself, my main concerns relate to the results shown in the PCA. For a more fully understandable genetic structure, I find that it is required to separate neutral

from non-neutral markers. I understand the authors not making this distinction as this was not the main objective of the study (and the amount of methods required increases, although perhaps an exploration using only BAYESCAN, or PCAdapt or any other selection-scanning software can be done without spending much time).

Our population genetic methods were meant to provide a brief justification for our species/strain assignments of these lineages, which can be difficult to distinguish morphologically. Sample sizes for this study were not large enough to thoroughly evaluate population structure. Our results are congruent with previous research on population structure of ciscoes (Bernal et al. 2022, Ackiss et al. 2021). In future work, we have sequenced hundreds of individuals from this group using low-coverage whole genome sequencing, which will be presented in other publications.

7. Sex of the individuals have not been reported, neither any signal that shows which markers are sexual (or if there is sexual chromosomes). Note that population structure results can change if not considering sex, and in this case, where the sample size is not big could bias the results substantially. For more information, see this DOI: [10.1111/mec.14217](https://doi.org/10.1111/mec.14217) and [10.1111/eva.12979](https://doi.org/10.1111/eva.12979).

Information on the sex of the samples in the study were not collected. There has not been any published evidence of sex chromosomes or sex-determining regions in Great Lakes *Coregonus* spp. to date. Salmonid sex chromosomes are not strongly differentiated, although there are sex determining genes that have been identified in wild populations of some salmon species (*sdY*; *Oncorhynchus* spp.) (Cavileer et al., 2015).

8. I have also a minor concern concerning the sampling scheme. Since the sample size is not big enough the choice of sampling coordinates may bias population structure results. However, I realise that given the low genetic differentiation between species, maybe this is a least concern.

This is a valid concern, however as stated, estimates of genetic differentiation (pairwise *Fst*) among Lake Superior *Coregonus* spp. from RAD sequencing are quite low (0.013 – 0.061; Ackiss et al. 2021).

REVIEWERS' COMMENTS:

Reviewer #1 (Remarks to the Author):

The authors made careful revisions in accordance with the three reviewers' comments. The present version of this manuscript is acceptable.

Reviewer #2 (Remarks to the Author):

This revised manuscript, which I reviewed before, has largely improved the quality of the presented reference genome. By adding data, especially HiC data, the reference genome presented now has a chromosome level assembly and hence matches state of art standards. The improvement of the reference genome also had a positive impact on the mapping statistics and quality of the re-sequencing data used to infer demographic histories.

The improved reference, some adjustment to the mapping parameters, and the masking of the reference genome before inferring demographic histories all had a positive impact on the results and manuscript. However, some of my previously raised concerns, which are more general to limitations of the approach, such as the unknown impact post divergence gene flow might have, or that divergences times are rather indirectly and descriptively inferred from N_e trajectories still remain. I think the analysis nevertheless provides interesting insights and the interpretation of results given in the manuscript can well be followed; however, I think limitations should be more clearly acknowledged and some interpretations of the results would better be phrased a bit more carefully and not as affirmative.

Further, I think it would be valuable to discuss the results (specifically the suggestion that the *Coregonus* species diverged before the LGM) in the context of what is known about the phylogeographic relationships of *Coregonus* species across the Great Lakes. I also would be curious to hear some thoughts regarding the observed pattern that *Coregonus* species radiated while their populations declined.

The manuscript clearly states a timeframe suggested for the divergence between the *Coregonus* species but misses a clear statement on when *Salvelinus* diverged. This might reflect the challenge to infer a time of divergence from such N_e trajectories.

Figure 2:

It might be more interesting to show the shape of the ice margins before the LGM specifically for the time when the radiation supposedly occurred. Why not highlight MIS5 as it is the suggested time of *Coregonus* divergence, and maybe do the same for the suggested time of *Salvelinus* divergence.

Line 136: Maybe start with a brief sentence giving some biological detail about the sample and species.

Line 152: Also here start with some biological details. Might be good to remind the reader that three species are sympatric and one is from another lake.

Line 190ff: Would it not be most relevant to point out the timespan when the N_e trajectories of the two species diverge?

Line 236 : the the

Line 238: earlier might be confusing here.

Line 279: why feedback loop, how did *Salvelinus* divergence feedback on the *Coregonus* species?????

Line 498 . missing

I could not access the data nor the github page.

REVIEWERS' COMMENTS:

Reviewer #1 (Remarks to the Author):

The authors made careful revisions in accordance with the three reviewers' comments. The present version of this manuscript is acceptable.

-Thank you!

Reviewer #2 (Remarks to the Author):

This revised manuscript, which I reviewed before, has largely improved the quality of the presented reference genome. By adding data, especially HiC data, the reference genome presented now has a chromosome level assembly and hence matches state of art standards. The improvement of the reference genome also had a positive impact on the mapping statistics and quality of the re-sequencing data used to infer demographic histories.

The improved reference, some adjustment to the mapping parameters, and the masking of the reference genome before inferring demographic histories all had a positive impact on the results and manuscript. However, some of my previously raised concerns, which are more general to limitations of the approach, such as the unknown impact post divergence gene flow might have, or that divergence times are rather indirectly and descriptively inferred from N_e trajectories still remain. I think the analysis nevertheless provides interesting insights and the interpretation of results given in the manuscript can well be followed; however, I think limitations should be more clearly acknowledged and some interpretations of the results would better be phrased a bit more carefully and not as affirmative.

-We have added another sentence in the discussion that states the limitations of the methods used: "We caution that estimation of divergence time through a visual examination of N_e trajectories could be confounded by other factors, such as by post-divergence gene flow." This adds to what we think are careful rhetorical hedges throughout the paper currently; our methods represent the current best approach available for addressing these questions, but we appreciate the reviewer's sentiment. We hope this caution is now effectively communicated.

Further, I think it would be valuable to discuss the results (specifically the suggestion that the *Coregonus* species diverged before the LGM) in the context of what is known about the phylogeographic relationships of *Coregonus* species across the Great Lakes. I also be curious to hear some thoughts regarding the observed pattern that *Coregonus* species radiated while their populations declined.

-To address the context of divergence and phylogenetic relationships we would need to create a phylogeny of Great Lakes cisco species. We have completed this with limited samples to show the distinctiveness of each species present in the study, but our results do not provide a proper phylogenetic analysis of the broader clade (genus *Coregonus*, subgenus *Leuciscus*). Previously published phylogenies of the *C. artedii* species complex in the Great Lakes used a limited set of genetic markers, which may be unreliable given the recent divergence between the species,

however we anticipate that a new flurry of phylogenetic studies in coregonines are on the near horizon, and will in part be potentiated by the genome assembly and SNP data we include here.

The manuscript clearly states a timeframe suggested for the divergence between the *Coregonus* species but misses a clear statement on when *Salvelinus* diverged. This might reflect the challenge to infer a time of divergence from such N_e trajectories.

-Yes, correct. We have deliberately avoided making a statement about the divergence of *S. namaycush* morphs in the Great Lakes. See comments

Figure 2:

It might be more interesting to show the shape of the ice margins before the LGM specifically for the time when the radiation supposedly occurred. Why not highlight MIS5 as it is the suggested time of *Coregonus* divergence, and maybe do the same for the suggested time of *Salvelinus* divergence.

-During our proposed time point of divergence (MIS5a), glacial extent was restricted to Northern latitudes. According to Batchelor et al. (2019; <https://www.nature.com/articles/s41467-019-11601-2>), the ice sheet would have been small enough to be well North of the Great Lakes catchment (Fig 1j from Batchelor et al. below). Therefore, adding a map to the timeline with these extents would not provide any additional information within the study area.

Line 136: Maybe start with a brief sentence giving some biological detail about the sample and species.

-A sentence was added describing the collection location and sex of the sample used as well as the tissue used for DNA extraction and sequencing.

Line 152: Also here start with some biological details. Might be good to remind the reader that three species are sympatric and one is from another lake.

-An addition to the first sentence was made to indicate the species and their respective populations (*i.e.* lake of origin) (line 167-168)

Line 190ff: Would it not be most relevant to point out the timespan when the Ne trajectories of the two species diverge?

-We agree that this would be a useful perspective to add, however there are some important differences between *Coregonus* and *Salvelinus* that complicate that approach and we instead decided to err on the side of caution. The *Coregonus* forms are currently recognized as distinct, nominal species, whereas the *Salvelinus* forms are still considered 'morphs' not yet far enough along the speciation continuum to warrant distinct names, suggesting that divergence time estimates might not be totally appropriate for those samples.

Line 236 : the the

-Deleted 'the' (line 257).

Line 238: earlier might be confusing here.

-The wording was adjusted to "... with the latter method nadir coinciding with the start of the LGM" (Line 259)

Line 279: why feedback loop, how did Savelinus divergence feedback on the Coregonus species?????

-Thank you for this comment. The reviewer makes an astute point - the *Salvelinus* divergence did not feedback on to *Coregonus* species. We have changed the wording to "evolutionary response". (Line 307)

Line 498 . Missing

-Done.

I could not access the data nor the github page.

-Github and NCBI data are now available; Dryad will be available on publication.